# Deep-Neural-Networks-Based Data-Driven Methods for Characterizing the Mechanical Behavior of Hydroxyl-Terminated Polyether Propellants

**DOI:** 10.3390/polym17050660

**Published:** 2025-02-28

**Authors:** Ruohan Han, Xiaolong Fu, Bei Qu, La Shi, Yuhang Liu

**Affiliations:** 1Xi’an Modern Chemistry Research Institute, Xi’an 710065, China; hanruohan204@163.com (R.H.); qubei630@163.com (B.Q.); shila_204@163.com (L.S.); 2Department of Geotechnical Engineering, Tongji University, Shanghai 200092, China; 2310037@tongji.edu.cn

**Keywords:** hydroxyl-terminated polyether (HTPE) propellant, mechanical behavior, data-driven, Bayesian optimization, deep neural network

## Abstract

Hydroxyl-terminated polyether (HTPE) propellants are attractive in the weapons materials and equipment industry for their insensitive properties. Storage, combustion, and explosion of solid propellants are affected by their mechanical properties, so accurate mechanical modeling is vital. In this study, deep neural networks are applied to model composite solid-propellant mechanical behavior for the first time. A data-driven framework incorporating a novel training–testing splitting strategy is proposed. By building Neural Networks (FFNNs), Kolmogorov–Arnold Networks (KANs) and Long Short-Term Memory (LSTM) networks and optimizing the model framework and parameters using a Bayesian optimization algorithm, the results show that the LSTM model predicts the stress–strain curve of HTPE propellant with an RMSE of 0.053 MPa, which is 62.7% and 48.5% higher than the FFNNs and the KANs, respectively. The R^2^ values of the LSTM model for the testing set exceed 0.99, which can effectively capture the effects of tensile rate and temperature changes on tensile strength, and accurately predict the yield point and the slope change of the stress–strain curve. Using the interpretable Shapley Additive Explanations (SHAP) method, fine-grained ammonium perchlorate (AP) can increase its tensile strength, and plasticizers can increase their elongation at break; this method provides an effective approach for HTPE propellant formulation.

## 1. Introduction

For weaponry development, insensitive propellants are an important direction and urgent need. The hydroxyl-terminated polyether (HTPE) propellant has passed all seven insensitive performance evaluation tests (Safe Drop, Fast Cook-off, Sympathetic Detonation, Fragment Impact, Bullet Impact, Slow Cook-off, and Jet Impact) of the U.S. Military Standard MIL-STD-2105C [1]. Polytetrahydrofuran and polyethylene glycol with low molecular mass are used to synthesize HTPE binder, and end-hydroxyl-blocked co-polyether is the matrix of HTPE propellant, which contains a solid-particulate oxidizer, a metal fuel, a plasticizer, and an auxiliary reagent [2]. In comparison to double-base propellants and other composite solid propellants, HTPE propellant has superior performance. Mechanical properties are superior over a wide temperature range. The excellent insensitivity properties of HTPE propellant make it a very attractive solid-propellant development direction, and it has a wide range of applications in rockets, missiles, and other weapons and equipment [3].

The mechanical properties of HTPE propellants affect their storage, combustion, and explosion. In the rocket launching process, the propellant needs to withstand a variety of complex loads, such as high temperature and high pressure, aerodynamic load, and inertial load, so it must have adequate strength, stiffness, and toughness to ensure the safety and success of the rocket launch [4].

Engineering and materials science use the stress–strain curve to determine key properties of materials, such as the elastic modulus, yield strength, ultimate strength, and elongation [5]. HTPE propellants exhibit complex mechanical behavior after being stressed and often undergo large displacements and strains during the deformation process, which makes it difficult to describe their mechanical properties [6].

Before conducting experiments, it is necessary to predict the properties of the composite propellant. Besides determining whether or not a particular need or application can be met, it reduces the cost of the experiment and reduces the likelihood of explosions. Three main types of stress–strain relationships are described for rubber materials. The Neo-Hookean model, the Kuhn–Grun model, the Arruda–Boyce model, and the Pucci–Saccomandi model are all models based on molecular theory [7], and phenomenological methods are based on the mechanics of the continuum medium, such as the Mooney–Rivlin model, the Ogden model, and the Yeoh model [8]. They are both superelastic models. Maxwell’s model [9], Kelvin’s model [10], and Schapery’s model [11] are all models based on viscoelastic theory [12]. Considering viscoelasticity, strain rate, peri-compression, and softening effects, Wubuliaisan M et al. developed a multiscale nonlinear viscoelastic constitutive model for high-elongation nitrate-plasticized polyether (NEPE) propellant [13]. The same year, Wubuliaisan. et al. [14] proposed a nonlinear viscoelastic constitutive model that took into account environmental factors and large deformation-induced damage to composite solid propellants. Li et al. [15] introduced the glassy and rubbery failure criteria into the viscoelastic constitutive model based on the strain energy density at ultra-low and ultra-high temperatures, respectively, establishing a unified model for the progressive damage and final failure of solid propellants. Zou et al. [16] proposed an eigenstructure model for viscoelastic polymerization to explain the mechanical behavior of composite solid-propellant particles/substrates. According to the visco-polymerization constitutive model, interfacial behavior exhibits a strong rate and temperature dependence. This method provides explanatory power, but modeling complex material systems is more challenging. Meanwhile, mechanical models are usually simplified in the modeling process by simplifying the problem or attaching assumptions that limit their ability to predict mechanical behavior. 

The rapid development of deep learning, also known as artificial neural networks (ANNs), can effectively overcome these difficulties by using open-source deep learning libraries and continuously optimized algorithms to model complexities involving complex nonlinear relationships between input and output variables in a relatively fast time, attracting significant attention in recent years for its ability to effectively model complex problems. To characterize the mechanical behavior of composite materials, several ANN-based methods have been proposed. Khoei et al. [17] developed an atomistic–continuum multiscale technique based on feedforward neural networks for simulating the mechanical behavior of composite materials, which can capture fracture and stress concentration phenomena. Khoei et al. [18] developed a multiscale computational framework based on the MLP-ANN architecture for modeling nanomaterial mechanical behavior with geometric and material nonlinearities. The method captures nonlinear material behavior with atomic accuracy and low computational cost. Kim et al. [19] presents a deep neural network that can be used to predict the mechanical behavior of composite materials. The predictions matched simulations and experiments well. Yuan et al. [20] proposed a self-tuning particle swarm algorithm to optimize an artificial neural network for predicting the mechanical behavior of rubber considering the coupling effect of rubber hardness and strain rate, which improved prediction accuracy by 26.5%. Without a physical model, ANN models can be constructed directly from stress–strain curves [21]. Recurrent neural networks (RNNs) with recurrent connections between hidden layers have been developed to overcome the difficulty of predicting mechanical behavior curves of composites [22]. However, RNNs face the problem of gradient explosion and gradient vanishing when learning long-term historical data [23]. Long Short-Term Memory (LSTM) networks (a type of RNN) have been developed to address this shortcoming [24]. In recent years, LSTM networks have been used to study composite material properties. Dorbane et al. [25] used LSTM and GRU models to predict stress–strain curves of aluminum alloy substrates at different temperatures. Under different temperature conditions, the LSTM and GRU models are able to capture the future trend of stress–strain curves well. Tanhadoust et al. [26] used DNN and LSTM recurrent neural networks to predict concrete behavior at elevated temperatures. Results indicate that the proposed LSTM model is reliable and effective in predicting concrete’s compressive strength, elastic model, and destructive strain. As machine learning continues to evolve and new machine learning models emerge, the Kolmogorov–Arnold Network (KAN) puts learnable activation functions on weights and approximates complex multidimensional functions by learning a series of one-dimensional functions, an algorithm for improving neural network accuracy and interpretability, as well as facilitating researchers’ discovery of new physical and mathematical concepts [27].

To date, machine learning techniques have been used to study composite solid-propellant performance [28,29,30,31]. There is still a gap in research on using machine learning to predict propellant mechanical properties. Complicated propellant compositions, variable mechanical behavior, and a lack of understanding of chemical and physical processes are the main challenges in predicting composite propellant performance. For predicting the mechanical behavior of propellants, machine learning and artificial intelligence approaches work in concert with experimental and empirical methods.

Overall, machine learning has shown tremendous potential in studying composite materials. Nevertheless, there are significant research gaps in predicting the mechanical behavior of composite solid propellants. To accurately predict complete stress–strain curves for HTPE propellants, a representative insensitive propellant, this study proposes a robust machine learning modeling strategy that combines deep neural networks with a novel training–testing split approach. Three types of propellants were selected for the study as case studies, including HTPE/AP/Al, HTPE/RDX/AP/Al, and HTPE/HMX/AP/Al. Models were constructed using Feedforward Neural Networks (FFNNs), Kolmogorov–Arnold networks (KANs), and Long Short-Term Memory (LSTM), trained using optimal hyperparameters obtained through Bayesian optimization, and their performance was evaluated. Furthermore, SHAP interpretable analysis was used to examine the effect of HTPE propellant components on its tensile strength and elongation at break. It has been demonstrated that the proposed models accurately predict the mechanical behavior of HTPE propellants.

## 2. Materials and Experiments

HTPE propellants and uniaxial tensile tests are described in this section. A hydroxyl-terminated polyether prepolymer (HTPE; Mn = 3310 g·mol^−1^, hydroxyl value is 0.604 mmol g^−1^), purchased from Liming Research & DesignInstitute of Chemistry Co., Ltd. (Luoyang, China), is used to prepare HTPE propellant samples, along with a curing agent, a polyfunctional isocyanate (N-100) with a 5.37 mmol·g^−1^ isocyanate group content, provided by Xi’an modern chemistry research institute (Xi’an, China), a plasticizer N-butyl-N-(2-nitroxy ethyl) nitramine (Bu-NENA), cyanoethylated polyamine (HX-878) as a bonding agent, which are from Tianyuan Aerospace Materials (Yingkou) Technology Co., Ltd. (Yingkou, China), and a catalyst to cure the cross-linking reaction. And the solid fillers were ammonium perchlorate (AP; particle sizes are 6–8 μm and 100–140 μm) and aluminum powder (Al; particle size is 5 μm), supplied by Xi’an modern chemistry research institute. 

To prepare HTPE propellants with different particle sizes, prepolymer, plasticizer, binder, TPB, Al, and AP were weighed in certain proportions; mixed in a mixer; and vacuumed to remove air bubbles. The particle grade information for HTPE propellant is shown in Table 1. After mixing HTPE propellant with N-100, a slurry was poured into a mold, evacuated in a vacuum chamber, degassed, and cured at 50 °C for seven days. Figure 1a shows the specific preparation process for HTPE propellant.

The cured samples were placed in a desiccator to dry for 24 h. The obtained propellants were sliced into 10 mm thick tablets with a slicing machine, and 120 × 25 × 10 mm dumbbell-type specimens were made in the sample making machine. After all the specimens were measured, they were placed in a heat preservation box and held at 25 °C for 1~2 h to ensure that the temperature inside and outside of each specimen remained uniform. The mechanical property test of HTPE propellant was carried out using a tensile test machine (AGS-J, Shimadzu, Kyoto, Japan), according to GJB770B-2005 [32] at a constant test temperature of 25 °C for the uniaxial tensile test of the specimen, and the loading rates were 0.5 mm/min, 2 mm/min, 10 mm/min, 20 mm/min, 50 mm/min, 100 mm/min, 200 mm/min, and 500 mm/min. The specimens were placed in the center of the testing machine. The material was first preloaded, and after preloading was completed, the material was stretched uniaxially at a constant rate until the sample broke. 

Each experiment was repeated three times, and since the results of the three experiments were relatively close, one set of data was selected for the following analysis. The propellant stress–strain curve, initial modulus, maximum tensile strength, and corresponding maximum elongation were obtained by processing the stress–strain curve. An illustrated schematic of HTPE propellant uniaxial tensile experiments is shown in Figure 1b.

## 3. Modeling Process

Deep neural networks can accurately predict rate-dependent stress–strain curves in complex nonlinear relationships. Data preprocessing is presented in this section, followed by the principles of feedforward neural networks and feedback neural networks in the category of deep neural networks. In this article, the model modeling theory is outlined, and the optimization principle is elaborated from two perspectives, namely, the optimization of the model structure and the optimization of the weights of the networks. Model development is discussed in relation to R^2^, RMSE, and MAE as evaluation indexes. 

### 3.1. Dataset Establishment and Preprocessing

Stress–strain curves for HTPE propellants were obtained from uniaxial tensile tests performed in our laboratory and the published literature. Data were obtained by stretching HTPE propellants at different temperatures and tensile rates. Table 2 summarizes the solid filler type, filler content, tensile temperature, tensile rate, and curve number. There are several variables included in this dataset, such as binder, plasticizer, curing agent, and crosslinker content for HTPE propellants; solid fillers containing incendiary agents (Al); oxidizer ammonium perchlorate (AP) in coarse and fine grains; high-energy explosives (RDX, HMX, and HATO); as well as mechanical test conditions (tensile temperature and tensile rate). The boxplots of the concentration trends and skewness of the data are shown in Figure 2. In HTPE propellant formulations, both the binder and solid filler content influence the mechanical properties. In solid fillers, Al content ranges from 5 wt% to 20 wt%, with a median of 10 wt%, whereas AP content for coarse particle size ranges from 17 wt% to 50 wt%, with a median of 27 wt%. The range of AP content for fine particle size is 8 wt%~28 wt%, and the median is 25 wt%. The HTPE content of adhesives varies from 5 wt% to 9.93 wt%, with a median of 8 wt%. The plasticizer content of adhesives varies from 10 wt% to 12.9 wt%, with a median of 11.8 wt%. The crosslinker content varies from 1.0 wt% to 2.75 wt%, with a median of 2.12 wt%. The selected input features are distributed within a reasonable range. This dataset contains representative stress–strain curves.

Preprocessing the data is the first step in developing a machine learning model to ensure that the data are evenly distributed, correctly formatted, and free of outliers. The model is trained, validated, and tested using the whole stress–strain curve instead of individual data points. A single curve divides the dataset into training and testing sets. The machine learning algorithms are biased toward the major categories with higher data density when the number of observations (data point density) on stress–strain curves is unequal or nearly equal for different parameter sets [34]. For each stress–strain curve, a polynomial fit was evaluated, and 50 equally spaced data points were regenerated on the best-fitting curve in place of the original curve. The categorical features were then converted to numerical features using median encoding (MEAN ENCODING) for training and predicting machine learning models. Due to the large number of dimensions of the original dataset, different types of variables have a wide range of values. In this case, the AP content (6–8 μm) ranges from 25% to 55%, while the tensile rate varies from 2 mm/min to 1000 mm/min. 

In the constitutive relationships, stress and strain are one-to-one correspondences [35]. Using the Spearman correlation coefficient (SCC), correlations were explored between parameters of material formulation, test temperature, tensile rate, strain, and stress [36]. In Figure 3, strain, tensile temperature, and loading rate are positively correlated with stress, and the Spearman correlation coefficients (SCCs) between strain, tensile temperature, and loading rate are 0.02, 0.11, and 0.18, respectively. According to the negative correlation between the AP of coarse particle size and Stress, and the positive correlation between the AP of fine particle size and Stress, the proportion of solid particles in the constitutive relationship of HTPE propellant materials should be considered. We selected Temperature, Tensile rate, AP content, Strain, and other formulation parameters as input layers and Stress as output layers. There are 12 inputs corresponding to one output in the data structure.

Models that use the original proportions of the dataset assign larger weights to features with larger ranges, which affects prediction results. A MinMaxScaler method from the Scikit-Learn library was used to transform the features in this study [37]. Normalization is calculated as Equation (1).(1)$$x_{new}=\frac{x-x_{min}}{x_{max}-x_{min}}$$
where *x* is the original value; $x_{new}$ is the normalized value; and $x_{min}$ and $x_{max}$ are the minimum and maximum values of the input features, respectively.

### 3.2. Deep Neural Network Models Used for Training

#### 3.2.1. Feedforward Neural Network

A feedforward neural network (FFNN) is a type of artificial neural network. Figure 4a shows its architecture. The trainable parameters are the weights and biases. Each node in one layer is connected to the nodes in the subsequent layer, but this network contains no loops or cycles. Each neuron in the hidden layer processes the input information through a set of connection weights and a fixed activation function to determine the output. Input data are propagated backward to the hidden layer where features are extracted and processed, and predictions are made based on the learned features [38]. The system is trained to recognize patterns by adjusting the connections between neurons [39].

#### 3.2.2. Kolmogorov–Arnold Network

Kolmogorov–Arnold Networks (KANs) [40], inspired by the Kolmogorov–Arnold representation theorem, serve as a viable alternative to feedforward neural networks. Figure 4b shows KAN’s architecture. KAN has a fully connected structure. FFNN places fixed activation functions on nodes (neurons), while KAN places learnable activation functions on edges (weights). The KAN model does not use linear weights, but instead uses univariate spline functions for each weight parameter. A KAN node simply sums incoming signals without transforming them in a nonlinear manner. Through the use of piecewise polynomial functions, spline functions can be used to model high-dimensional data as one-dimensional polynomial functions. As a result, KAN is able to capture more complex data relationships than FFNN. However, the spline functions have the curse of the dimensionality problem, which makes them unable to handle high-dimensional data effectively.

#### 3.2.3. LSTM Neural Network

The mechanical behavior of HTPE propellants depends on the loading path (history). The recurrent neural network (RNN) can learn directly from time-dependent data, where the nodes are chained together recursively following the evolution of the sequence [41]. Long Short-Term Memory (LSTM) networks are special forms of RNNs that add LSTM units on top of RNNs. LSTM architecture is shown in Appendix A. The LSTM controls information flow through forget gates, input gates, and output gates. The combination of an S-function (σ) and point-by-point multiplication (x) is a gate that selectively lets information through. The forget gate decides what information to discard, the input gate determines what to keep in memory, and the output gate determines what to output based on the cell’s memory and input. By utilizing LSTM units as gate control mechanisms, the gradient vanishing and gradient explosion problems of RNN can be effectively handled. In addition, it can effectively deal with the long-term dependence and local dependence of an RNN [42]. LSTM can memorize more previous information to obtain more accurate models, which proves that LSTM networks are an excellent choice for analyzing time series. An LSTM model with t time steps is shown in Figure 5. LSTM networks generate 3D networks based on LSTM cells along height, width, and time dimensions.

### 3.3. Model Optimization Methods

Hyperparameter tuning is an important step before machine learning model training to find the optimal hyperparameters for a neural network. The optimal set of hyperparameters can be determined by training and evaluating the model with different sets of hyperparameters [43]. Currently, grid search methods require a lot of time and computation for random searches. Since overly complex models are prone to overfitting, and overly simple models do not provide enough prediction, choosing the right hyperparametric optimization algorithm is therefore crucial. Based on the currently tested hyperparameter combination, the Bayesian optimization algorithm predicts the next possible hyperparameter combination that will offer the greatest benefit [44]. Previous studies have shown that Bayesian optimization is effective [45,46].

In deep neural networks, the error function is used as a performance metric to obtain the optimal weights and bias values; therefore, in order to make the network optimal, it is necessary to make the error function as small as possible [47]. During training, Limited-BFGS, SGD, and Adam methods are used to minimize the loss function. 

### 3.4. Modeling Process of Deep Neural Networks

A deep learning approach represents mechanical behavior based on stress and strain data. HTPE propellant intrinsic laws are stored in the weights of all links of the model as a result of stress–strain data collected from tensile experiments. Based on the DNN framework, Figure 6 shows the modeling process for predicting the mechanical behavior of HTPE propellant. The modeling process is divided into three stages: stress–strain curve collection and preprocessing, Bayesian hyperparameter optimization to determine the network architecture, and weight optimization of the determined network. Ultimately, the optimal neural network structure is determined by predicting the mechanical behavior of HTPE propellants. 

### 3.5. Evaluation Metrics of DNN Models

To characterize and compare the prediction accuracy of the three neural network models, the coefficient of determination (R^2^), root mean square error (RMSE), and mean absolute error (MAE) evaluation metrics are used. Based on these metrics, we can determine how well the predicted values match the actual values. 

A coefficient of determination (R^2^) is defined as the ratio of the sum of squares of the residuals to the total sum of squares and relates the regression model to observed data differences. The R^2^ metric ranges from 0 to 1. The closer it is to 1, the more accurate the model’s prediction is, and the more adequately it explains the true value [48]. Root Mean Square Error (RMSE) measures the predictive accuracy of a regression model by assessing the difference between predicted and true values. Outliers can be detected very easily, so it is an excellent indicator of prediction accuracy. The smaller the root mean square error, the more accurate the prediction [49]. Mean absolute error (MAE) measures the average difference between the predicted and actual values. It uses the decision value of the error, so it is more stable in handling outliers. The smaller MAE, the better the model’s performance [50]. Table 3 shows the formula for the evaluation metrics.

## 4. Result and Discussion

In this section, the stress–strain relationships are first analyzed for different formulations of HTPE propellant at different loading rates. For DNN, the optimal network structure, optimizer, number of iterations, and learning rate are selected. FFNN, KAN, and LSTM models are compared for prediction accuracy to determine the superiority of the proposed model. Finally, SHAP interpretable analysis is used to analyze the effects of different formulation components on propellant mechanical properties.

### 4.1. Results of Uniaxial Tensile Test of HTPE Propellant

HTPE propellants at different stretching rates are shown in Figure 7 in terms of stress–strain relationships. This is a typical nonlinear phenomenon. The ultimate stress and strain-at-break of the propellant increases with the loading rate, and its Young’s modulus increases as well [51].

Figure 7a,c demonstrate that the stress–strain relationship of HTPE propellant exhibits quasi-linear characteristics when loaded at a low tensile rate. The stress–strain relationship of HTPE propellant shows obvious nonlinear characteristics when strain exceeds 0.07. When the tensile rate increases, the propellant tends to fracture, and the elastic modulus, tensile strength, and elongation all rise. Increasing strain rates cause the curled polymer chains inside the propellant to not be able to stretch in time, and their tensile movement cannot keep up with a change in tensile strain [52]. According to Figure 7b,d, the stress–strain curves for the propellant demonstrate quasi-linear characteristics at the initial loading stage at various higher tensile rates. At 100 mm/min, yielding and stress softening phenomena are observed, and at higher tensile rates, yielding and phenomenal strain softening are more evident. The time taken for chain segments to move and rearrange within a polymer is compressed, and the molecular chain movement and rearrangement ability of the material is significantly diminished. Thus, the local stress concentration phenomenon is more pronounced, leading to a more pronounced yielding and strain softening response [53]. At low strain rates, HTPE propellant typically undergoes ductile failure. However, as the strain rate increases, the failure mode gradually transitions to brittle failure. When the stress reaches a critical value, the material will fracture suddenly with almost no plastic deformation.

The AP of fine particle size of formulation 2 is 5% higher than that of formulation 1 at constant solid filler content. At 500 mm/min, formulation 2 reaches a maximum tensile strength of 0.64 MPa. Due to the increase in fine particle size filler, the macromolecules are able to adsorb into the fine particles, creating a physical crosslinking point, increasing the material’s mechanical strength [54].

### 4.2. Neural Network Architecture Determination

Before constructing models to predict HTPE propellant mechanical behavior, the effect of the number of data points, fitted through polynomial approximation, on the accuracy of stress–strain curve predictions was analyzed with computational efficiency in mind. During the dataset preprocessing stage (Section 3.1), polynomial fitting was used to generate curves with varying numbers of data points. The number of data points has a significant impact on the root mean square error (RMSE) of the models on both the training and testing sets, as shown in Appendix A. By increasing the number of data points from 10 to 30, the RMSE decreases significantly. Beyond 50 data points, the RMSE becomes stable with little further improvement. To balance prediction accuracy and computational efficiency, this study used 50 evenly spaced data points along the strain axis to represent each curve.

To construct the best model for predicting the mechanical behavior of HTPE propellant, Bayesian optimization algorithms are used to optimize the hyperparameters of FFNN, KAN, and LSTM. In FFNN and KAN, the number of hidden layers and the number of neurons in each layer are expressed, while in LSTM networks, the number of hidden layers is expressed, as well as the number of LSTM hidden layers, as well as the number of nodes in each layer. Since the LSTM model deals with time series data, non-time series data (compositional formulation of HTPE propellant, etc.) were copied in multiple copies to match the time step of the Strain time series data, and the non-time series data were combined with the time series data (Strain) and fed into the LSTM model for training, which was arranged as follows: (Samples, Time step, Features) with five steps set. Generally, neural network hyperparameters are divided into three categories: network parameters, optimization parameters, and regularization parameters. The network parameters determine how the network layers interact, including the number of neurons, the number of layers, and the activation function. Learning rate, optimizers, and loss function are all optimization parameters. Regularization includes weight decay coefficients and discard rates. According to the modeling strategy proposed in Section 3.4, Optuna, an optimization framework based on the Bayesian optimization algorithm, is used to implement the hyperparameter optimization of the model. Table 4 shows the final hyperparameter optimization results of the three architectures after 150 iterations of Optuna. As shown in Figure 8, FFNN, BPNN, and LSTM parallel coordinate plots under different architectures reflect the hyperparameter optimization process map for the model, a visual representation of the hyperparameter optimization process. This trend allows us to observe how different model architecture parameters affect model prediction accuracy, leading to a narrowing of the search space and an improvement in computational efficiency.

After determining the hyperparameters, the changes in the models’ prediction results during training were visually analyzed using the stress–strain curve of the HTPE/RDX/AP/Al propellant as an example. Figure 9 shows the predicted results at different iteration epochs compared with the experimental data. The study found that as the number of training epochs increased, the predictions gradually approached their actual values. Ultimately, after 10,000 epochs, the predicted curve, composed of 50 data points, almost completely overlapped with the actual stress–strain curve.

### 4.3. Comparison of Models with Different Architectures

The stress–strain curves of different types of HTPE propellants are characterized using the FFNN, KAN, and LSTM models. As shown in Figure 10, the three neural network methods predict stress–strain curves of HTPE/AP/Al and HTPE/HMX/AP/Al propellants at the same temperature with different tensile rates, as well as predicted stress–strain curves for the same tensile rate at different temperatures for HTPE/RDX/AP/Al propellants. Both KAN and LSTM models can accurately predict materials’ stress–strain curves compared to the FFNN model. In terms of prediction results, the LSTM model significantly outperforms the FFNN model. The LSTM model demonstrated superior performance in capturing the changes in stress during the initial loading phase, particularly in the elastic region (at smaller strains). Additionally, LSTM identified the strain range and corresponding stresses at the yield point, which has important implications for propellant performance studies.

In Table 5, we present the R^2^, RMSE, and MAE metrics for the HTPE/AP/Al, HTPE/RDX/AP/Al, and HTPE/HMX/AP/Al propellants on the training and test sets so that the prediction effectiveness of the LSTM, FFNN, and KAN models can be evaluated. 

On the training and test datasets for HTPE/RDX/AP/Al propellants, the average MAE values of the LSTM model are 0.0103 and 0.0432, respectively, which are significantly better than those of FFNN (MAE of 0.0997 MPa) and KAN (MAE of 0.0877 MPa). In addition, both the training and test datasets showed LSTM models with R^2^ values greater than 0.99, while FFNN and KAN had R2 values below 0.97. Prediction results were similar for HTPE/AP/Al and HTPE/HMX/AP/Al propellants. As a result of the LSTM model having the smallest MAE and RMSE and the highest regression coefficient (R^2^), the model consistently provided excellent accuracy during training. KAN ranked second, while FFNN performed poorly.

In order to compare the errors of the three prediction models throughout the prediction process, three propellants from the training set and the test set were selected. Equation (2) illustrates the error E.(2)$$E=\left|\tilde{y_{i}}-y_{i}\right|$$
where E is the prediction error. Comparing the errors of the three neural network models throughout the prediction process can be done with Figure 11. During the whole loading process, the prediction errors of the three models are larger at the beginning and the end. At the beginning and end of loading, there is only one side of the data. The LSTM model error is lower than FFNN and KAN during loading.

Based on Figure 12 and Figure 13, the proposed model successfully predicts stress–strain relationships for HTPE propellants. Training curves are shown in blue, while test curves are shown in red dashed lines. Predicted values and experimental values show a high level of agreement.

In Figure 12, the LSTM model predicts the stress–strain curves of HTPE/AP/Al and HTPE/HMX/AP/Al propellants under different tensile rates (100 mm/min and 10 mm/min). High-tensile-rate (100 mm/min) and low-tensile-rate (10 mm/min) curves are within reasonable ranges, achieving R^2^ values of 0.99 and 0.98, respectively. The results indicate that the LSTM model accurately captures trends in tensile strength, elastic modulus, and peak strain under varying tensile rates. In particular, the model successfully captures the increase in tensile strength at high tensile rates. This is closely related to the insufficient relaxation of molecular chains and enhanced structural rigidity at high rates. 

Figure 13 demonstrates the stress–strain curve predictions of the LSTM model for HTPE/RDX/AP/Al propellant at different test temperatures and tensile rates (100 mm/min). The R^2^ value of the model reaches 0.993, showing a high degree of consistency with the experimental results. Meanwhile, the LSTM model captured the trend of decreasing tensile strength with increasing temperature. This is due to the decrease in peak load and the weakening of specimen stiffness. In addition, the model reproduced the change in the upward slope of the stress–strain curve and its increase with increasing tensile rate.

In Figure 14, the LSTM model predicts the stress–strain curves of HTPE/HATO/AP/Al propellants under different tensile temperatures and different coarse-particle-sized AP contents, with an R^2^ value reaching 0.998. The LSTM model captures the influence trends of the coarse-particle-sized AP content on the tensile strength and elastic modulus at different test temperatures. Especially in a low-temperature environment, the model reveals that both the tensile strength and the elastic modulus increase with the increase in the coarse-particle-sized AP content, but this increase effect gradually diminishes. At low temperatures, the mobility of molecular segments in the elastomer matrix is significantly weakened, and the coarse-particle-sized AP undertakes more tensile loads, enhancing the material’s rigidity. However, an increase in the content of coarse-particle-sized AP can lead to uneven distribution and aggravated stress concentration, restricting the enhancement effect and causing the increase in tensile strength and elastic modulus to gradually decrease as the content of coarse-particle-sized AP increases.

In summary, the visual comparison of the experimental results and the predictions of the model shows that the LSTM model can predict the stress–strain relationship of HTPE propellant under different conditions with high accuracy, which provides a reliable tool for the study of propellant performance.

### 4.4. Outcomes of SHAP Explanation Analysis

In order to investigate which components of the HTPE propellant contribute to the mechanical properties of the HTPE propellant, the research objective is to extract two representative mechanical properties, namely, tensile strength and elongation at break, from all the stress–strain curves. The SHAP method is used to perform an interpretability analysis from both a global and a local perspective using the already trained optimal predictive model for HTPE propellant properties [55].

By assessing the importance of features in terms of global interpretation, the SHAP algorithm evaluates the influence of features on algorithm predictions. In Figure 15a and Figure 16a, the SHAP values (on the *x*-axis) indicate the contribution degree for each data point. The vertical axis (*y*-axis) represents different features. Each data point represents a SHAP value for each sample under that feature, with redder colors indicating higher SHAP values, and bluer colors indicating lower SHAP values. Based on the swarm plots, both the high-energy explosives and the coarse-particle-sized AP have negative effects on tensile strength and elongation at break. The lower their contents, the smaller the tensile strength. It is because the particle size of high-energy explosives is generally 100 μm and above, and the coarse-particle-sized AP is around 100–140 μm, which is more prone to debonding. AP for fine particles has a positive influence on tensile strength. The plasticizer promotes the elongation at break of the propellant but reduces tensile strength. A higher plasticizer content results in a greater elongation at break and a smaller tensile strength. Figure 15b and Figure 16b show the ranking of features’ importance, and it is evident that solid fillers have a greater impact on mechanical properties than the binder matrix, but usually negatively. 

Additionally, SHAP provides local interpretability analysis, which visualizes and analyzes the influence of sample features on mechanical properties. In Table 6, HTPE/AP/Al propellant characteristics and corresponding SHAP values are presented. A visualization of the model output can be seen in Figure 17. The figure shows how the model derives the final predicted value from the baseline value by visualizing the contribution of each feature. Blue bars indicate a negative influence on the result, and red bars indicate a positive influence. The baseline value is calculated based on the average of the original model’s predicted values in the training data, which is 1.135. Based on the sample features, the final model output is deviated from the baseline value by a certain amount. The coarse-particle-sized AP is the attribute that has the greatest influence on the tensile strength in this sample, further reducing the baseline value by 0.736. According to the results, the sample has a positive effect on the HTPE propellant under these conditions. The SHAP values can be used to conduct targeted formulation research on HTPE propellants.

Based on the comprehensive analysis results above, AP with fine particle size and solid filler content can correspondingly enhance the tensile strength of the HTPE propellant. The total solid fillers content, the plasticizer content, and the HTPE content all contribute to the elongation of the HTPE propellant at break. This is of considerable importance for designing HTPE propellants with high tensile strength and elongation at break.

## 5. Conclusions

In this study, deep neural networks are applied for the first time to explore the constitutive relationship of composite solid propellants based on uniaxial tensile experiments of HTPE propellants, and a data-driven framework combining deep neural networks with a new training–testing splitting strategy is developed to characterize the mechanical behaviors of HTPE propellants based on their formulation components and mechanical experimental conditions. Laboratory experiments were first conducted to investigate the uniaxial tensile stretching behavior of different formulations of HTPE propellants at varying tensile rates. These experiments focused on strength, elasticity modulus, maximum stress, strain at break, and stress–strain curves. Subsequently, based on the combination of new data-splitting strategies, FFNN, KAN, and LSTM models were developed, and the optimal framework of the models as well as the model parameters were determined by a Bayesian optimization algorithm in an attempt to predict the complete stress–strain relationship of HTPE propellants under different formulations and test conditions. The accuracy of the proposed framework is validated through experimental studies. The LSTM model significantly improves model fit and new data prediction. For the HTPE/RDX/AP/Al propellants, the LSTM model reduces the RMSE from 0.198 MPa (FFNN) to 0.0684 MPa (LSTM), with an improvement in error accuracy of 62.7%. The R^2^ values of LSTM are over 0.99 for both the training set and the test set, whereas the R^2^ values of FFNN and KAN for the test set are 0.961 and 0.967, respectively. HTPE/AP/Al propellants, HTPE/HMX/AP/Al propellants, and HTPE/HATO/AP/Al propellants showed similar prediction results. LSTM successfully captures the trend of increasing tensile strength at high tensile rates and decreasing tensile strength at higher tensile temperatures. The LSTM model captures the trends of how the content of coarse-sized AP affects the tensile strength and elastic modulus at different test temperatures. LSTM also captures the yield point in the curve and its associated stress magnitude more efficiently. The strategy accurately predicts the mechanical behavior of materials with limited data. It can analyze complex composition–property relationships and reduces the need for expensive, hazardous, and time-consuming physical experiments. It has significant applications in the study of the formulation and performance of composite solid propellants. 

## Figures and Tables

**Figure 1 polymers-17-00660-f001:**
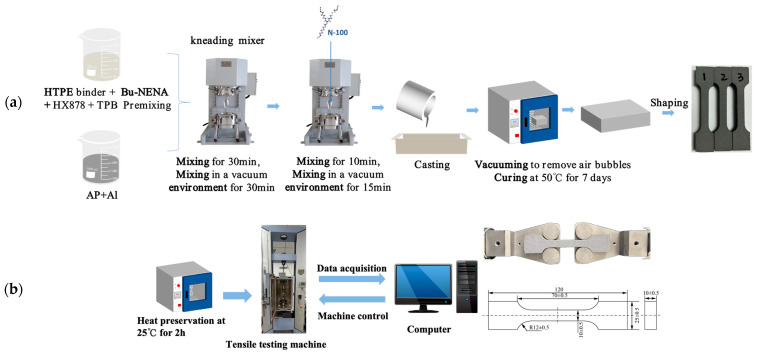
(**a**) HTPE propellant preparation process. (**b**) Schematic diagram of HTPE propellant uniaxial tensile experiment and specimen (unit: mm).

**Figure 2 polymers-17-00660-f002:**
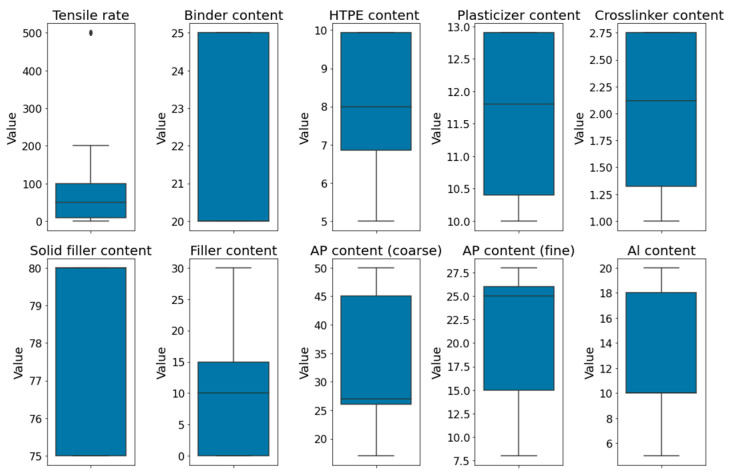
Boxplot of the selected input numerical features.

**Figure 3 polymers-17-00660-f003:**
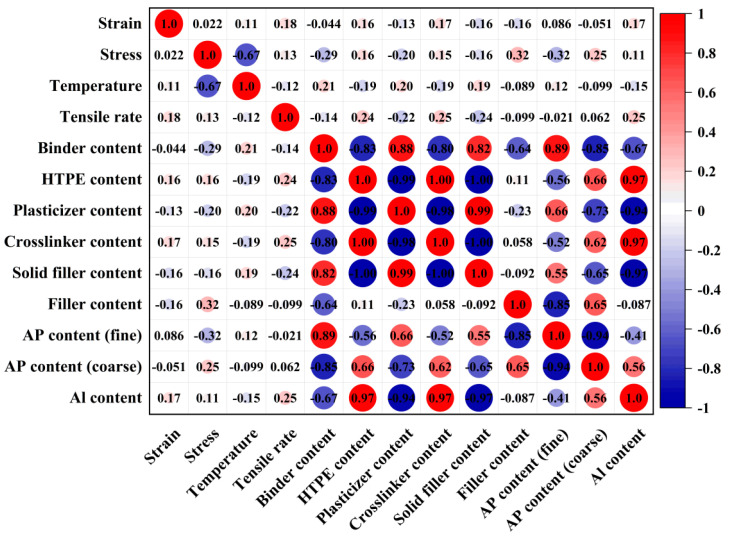
Heatmap of the SCC of each pair in the original 13 features. Red and blue colors represent positive and negative correlations, respectively.

**Figure 4 polymers-17-00660-f004:**
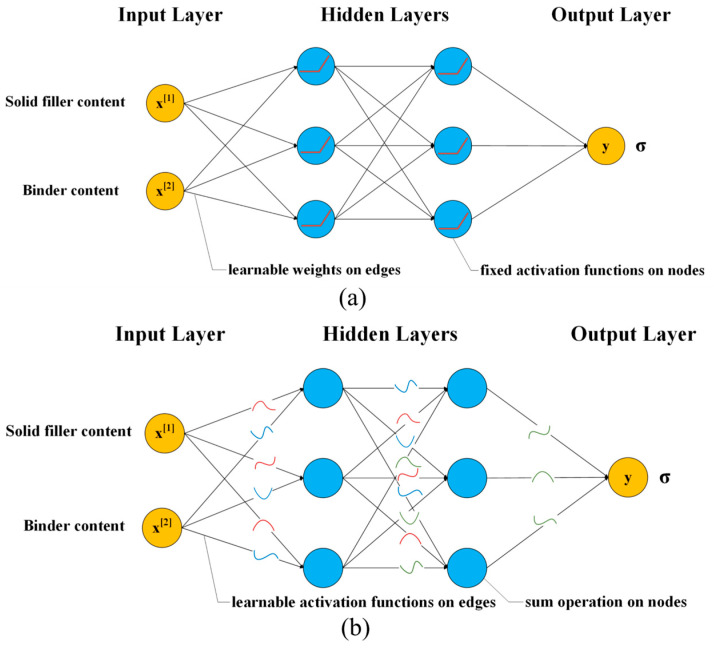
Frameworks of (**a**) Feedforward artificial neural networks and (**b**) Kolmogorov–Arnold networks.

**Figure 5 polymers-17-00660-f005:**
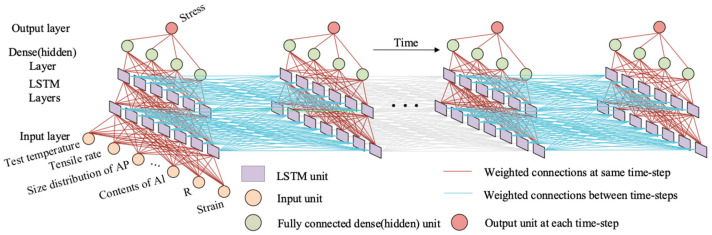
Three-dimensional architecture of LSTM deep neural network.

**Figure 6 polymers-17-00660-f006:**
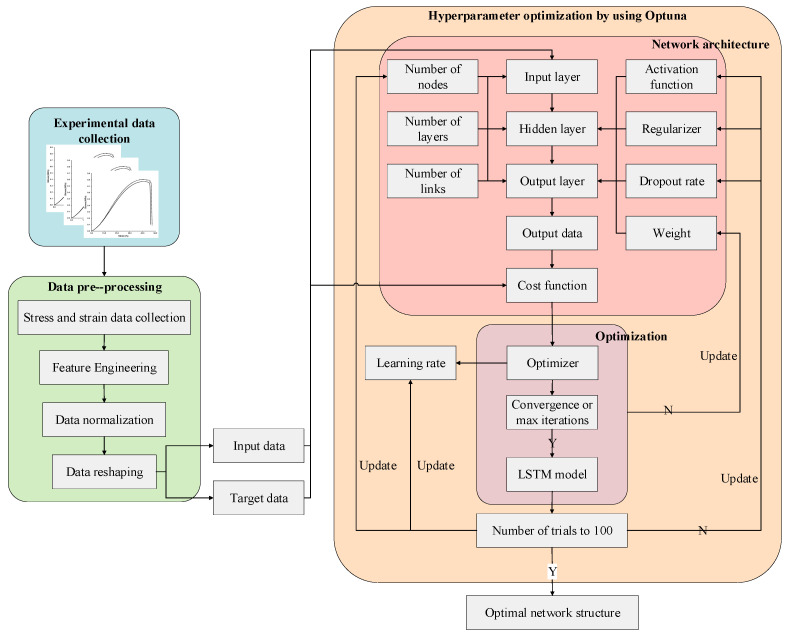
Diagram of the DNN modeling framework for predicting the mechanical behavior of HTPE propellant.

**Figure 7 polymers-17-00660-f007:**
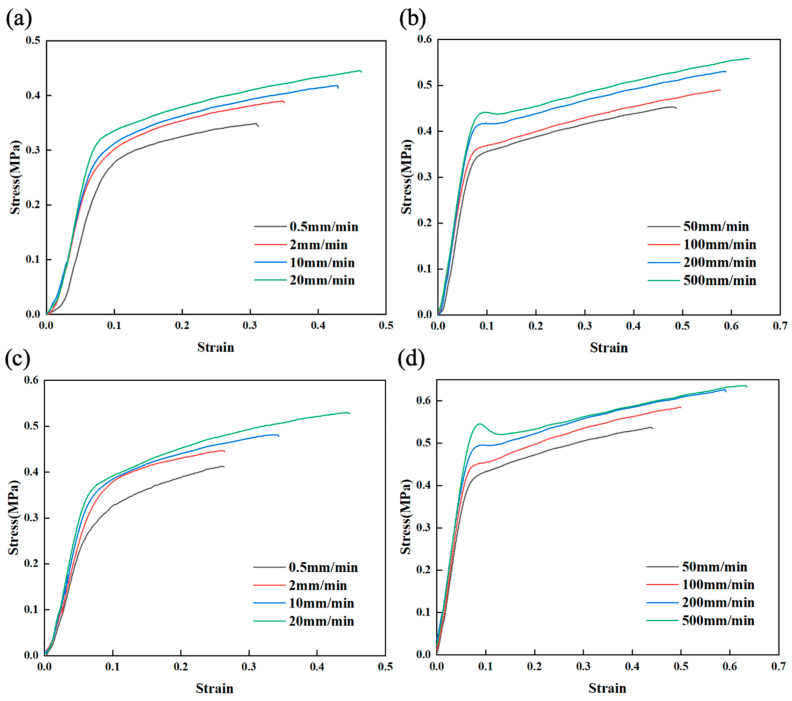
Stress–strain curves of formulation 1 under different (**a**) low and (**b**) high tensile rates. Stress–strain curves of formulation 2 under different (**c**) low and (**d**) high tensile rates.

**Figure 8 polymers-17-00660-f008:**
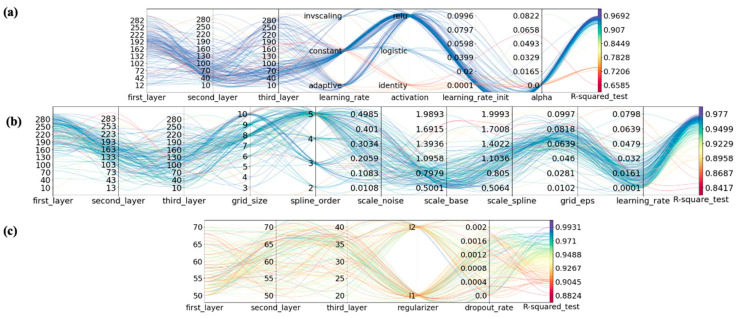
Parallel coordinate plot of hyperparameter optimization of (**a**) MLP, (**b**) KAN, and (**c**) LSTM.

**Figure 9 polymers-17-00660-f009:**
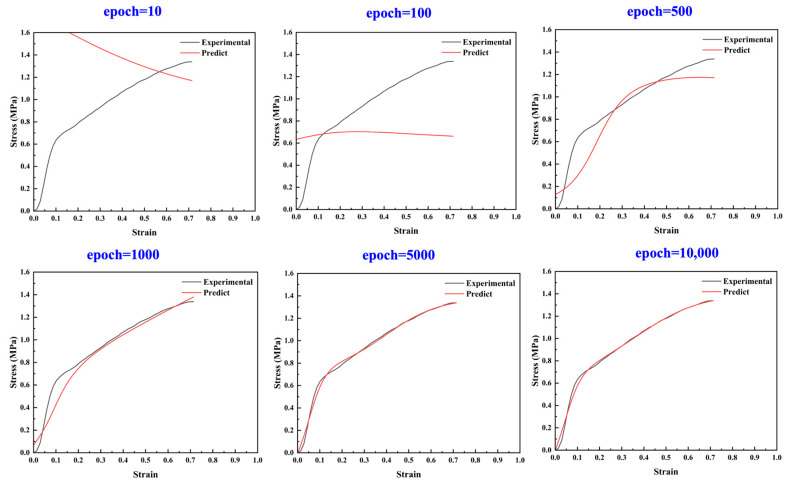
Neural network model training in process predicted stress–strain curves with the number of model iterations.

**Figure 10 polymers-17-00660-f010:**
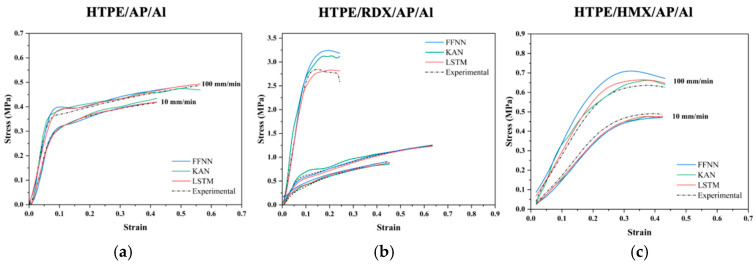
Stress–strain curves of (**a**) HTPE/AP/Al, (**b**) HTPE/RDX/AP/Al, and (**c**) HTPE/HMX/AP/Al propellants predicted by FFNN, KAN, and LSTM models.

**Figure 11 polymers-17-00660-f011:**
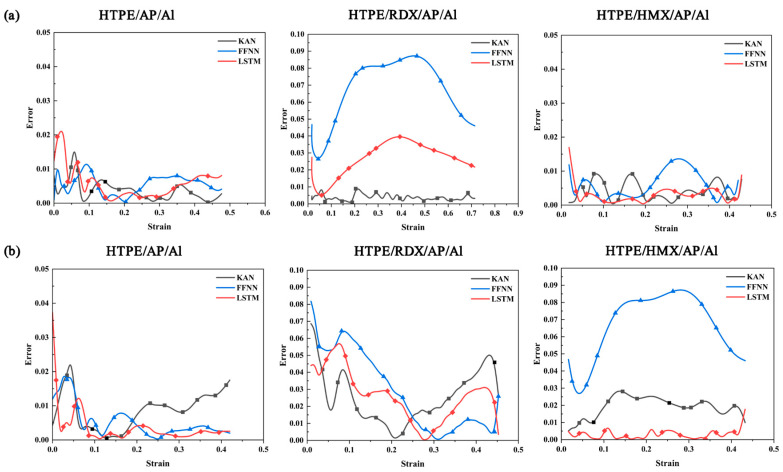
Comparison of the errors of FFNN, KAN, and LSTM on the prediction results of the (**a**) training set and (**b**) test set.

**Figure 12 polymers-17-00660-f012:**
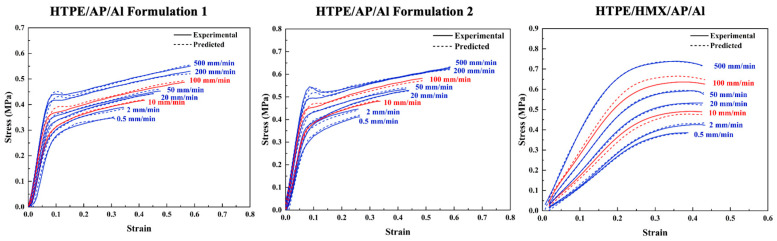
Results of using LSTM model to predict stress–strain curves of HTPE/AP/Al and HTPE/HMX/AP/Al propellants at different tensile rates.

**Figure 13 polymers-17-00660-f013:**
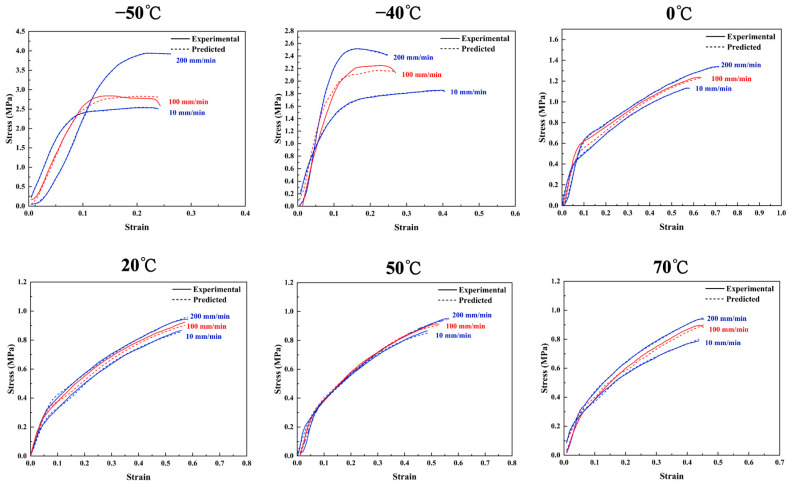
Results of using LSTM model to predict stress–strain curves for HTPE/RDX/AP/Al propellants at different test temperatures and tensile rates.

**Figure 14 polymers-17-00660-f014:**
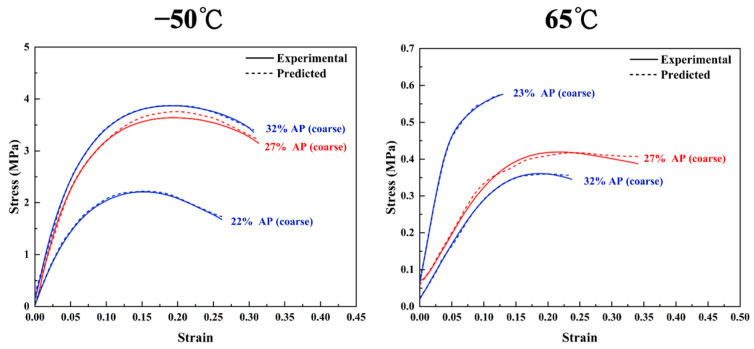
Results of using LSTM models to predict stress–strain curves for HTPE/HATO/AP/Al propellants at different test temperatures for different content of coarse-particle-sized AP.

**Figure 15 polymers-17-00660-f015:**
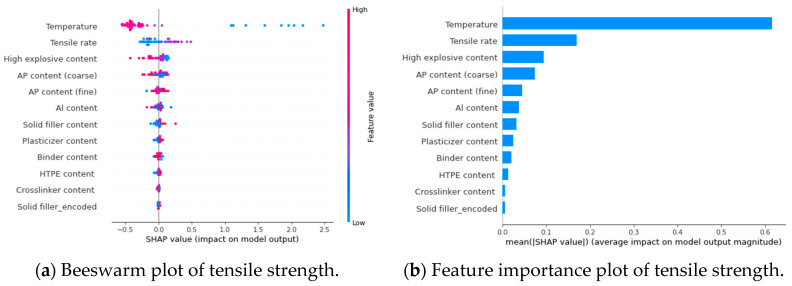
Global interpretations analysis by SHAP values for the input features of tensile strength.

**Figure 16 polymers-17-00660-f016:**
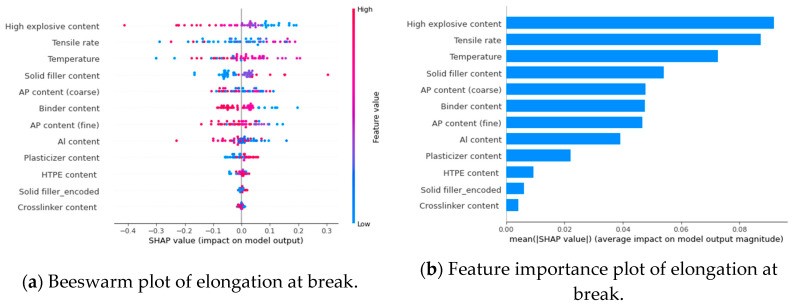
Global interpretations analysis by SHAP values for the input features of elongation at break.

**Figure 17 polymers-17-00660-f017:**
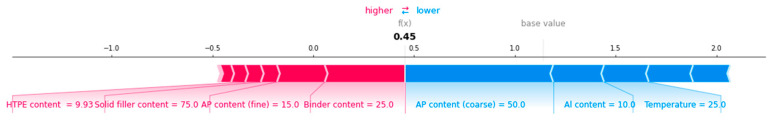
Individual interpretations for the maximum tensile strength of HTPE/AP/Al propellant of formulation 1 at 100 mm/min.

**Table 1 polymers-17-00660-t001:** HTPE propellant formulation information.

Name	HTPE	Bu-NENA	N-100	Al	AP (6–8 μm)	AP (100–140 μm)
Formulation 1	9.93%	12.9%	2.02%	10%	15%	50%
Formulation 2	9.93%	12.9%	2.02%	10%	20%	45%

**Table 2 polymers-17-00660-t002:** Detailed information on HTPE propellant collected.

Solid Filler Classification	Solid Filler Content (wt.%)	Temperature (°C)	Tensile Rate (mm/min)	Number of Curves	Producer
HTPE/AP/Al	75	25	0.5, 2, 10, 20, 50, 100, 200, 500	26	Our group
HTPE/HATO/AP/Al	77~80	−50, 20, 65	2, 100	12	Our group
HTPE/HMX/AP/Al	80	20	0.5, 2, 10, 20, 50, 100, 500	7	Our group
HTPE/RDX/AP/Al	80	−50, −40, 0, 20, 50, 70	10, 100, 200	18	[33]

**Table 3 polymers-17-00660-t003:** Model evaluation metrics.

Metric	Equation
R^2^	$R^{2}=1-\frac{\sum_{i=1}^{m}{\left(\tilde{y_{i}}-\bar{y_{i}}\right)}^{2}}{\sum_{i=1}^{m}{\left(y_{i}-\bar{y_{i}}\right)}^{2}}$
RMSE	$RMSE=\frac{\sqrt{\sum_{i=1}^{m}{\left(y_{i}-y_{i}\right)}^{2}}}{m}$
MAE	$MAE=\frac{1}{m}\sum_{i=1}^{m}\left|y_{i}-\tilde{y_{i}}\right|$

where m is the total number of samples, $y_{i}$ is the actual value of the samples, $\tilde{y_{i}}$ is the predicted value of the samples, and $\bar{y_{i}}\;$ is the average of the actual values of the samples.

**Table 4 polymers-17-00660-t004:** Optimal neural network architecture for FFNN, KAN, and LSTM models.

Model	Bayesian Hyperparametric Optimization Results for Predicting Stress–Strain Curves of HTPE Propellant
FFNN	first_layer: 244; second_layer: 177; third_layer: 3; activation: logistic; solver: L-BFGS; learning_rate: invscaling; alpha: 0.000362764; learning_rate_init: 0.007686944
KAN	first_layer: 285; second_layer: 152; third_layer: 102; grid_size: 9; spline_order: 5; scale_noise: 0.2819; scale_base: 0.5297; scale_spline: 1.1036; grid_eps: 0.0793; learning_rate: 0.0177
LSTM	first_layer: 54; second_layer: 66; third_layer: 36; solver: Adamx; regularizer: l2; dropout_rate: 0.00000479

**Table 5 polymers-17-00660-t005:** Performance summary of three models for predicting the mechanical behavior of HTPE propellants.

HTPE Propellants	Model	R^2^		RMSE		MAE	
		Training	Testing	Training	Testing	Training	Testing
HTPE/AP/Al	FFNN (Limited-BFGS)	0.994	0.992	0.011	0.012	0.008	0.009
	KAN (Adam)	0.997	0.993	0.0068	0.0104	0.0047	0.008
	LSTM (SGD)	0.996	0.995	0.0088	0.0086	0.0059	0.0064
HTPE/RDX/AP/Al	FFNN (Limited-BFGS)	0.999	0.9608	0.012	0.198	0.008	0.0997
	KAN (Adam)	0.999	0.967	0.0059	0.1442	0.0042	0.0877
	LSTM (SGD)	0.999	0.993	0.0147	0.0684	0.0103	0.0432
HTPE/HMX/AP/Al	FFNN (Limited-BFGS)	0.997	0.918	0.011	0.052	0.008	0.046
	KAN(Adam)	0.999	0.983	0.0052	0.0231	0.0041	0.026
	LSTM (SGD)	0.999	0.983	0.005	0.024	0.004	0.022
HTPE/HATO/AP/Al	FFNN (Limited-BFGS)	0.999	0.972	0.016	0.25	0.01	0.187
	KAN (Adam)	0.999	0.979	0.004	0.216	0.002	0.165
	LSTM (SGD)	0.999	0.998	0.019	0.061	0.012	0.048
Total	FFNN (Limited-BFGS)	0.999	0.969	0.012	0.142	0.008	0.06
	KAN (Adam)	0.999	0.977	0.006	0.103	0.004	0.05
	LSTM (SGD)	0.998	0.993	0.014	0.053	0.009	0.032

**Table 6 polymers-17-00660-t006:** Details of the tensile strength of HTPE/AP/Al propellant of formulation 1 at 100 mm/min.

Feature	HTPE Content	Solid Filler Content	AP Content (Fine)	Binder Content	AP Content (Coarse)	Al Content	Temperature
Values	9.93%	75%	15%	25%	50%	10%	25 °C
SHAP Values	0.077	0.08	0.237	0.397	−0.736	−0.253	−0.222

## Data Availability

Data available on request due to privacy.

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
