# Peer review of "Deep-Neural-Networks-Based Data-Driven Methods for Characterizing the Mechanical Behavior of Hydroxyl-Terminated Polyether Propellants"

_polymers, 2025, doi:10.3390/polym17050660_

Round 1

Reviewer 1 Report

Comments and Suggestions for Authors

This article presents a methodology for numerically predicting the mechanical behavior of polymer-based propellants using neural networks. Experimental test data was curated in the laboratory and also extracted from the literature. The methodology is quite extensively explained and validated. The results have been shown to be quite robust that able to predict variation in stress-strain behaviour with changing composition, temperature and other parameters. 

1) An interesting phenomenon is visible with an increasing strain rate that shows necking in the test zone of the specimen while at a low strain rate, a brittle failure seems to happen. Can authors comment on the failure mode whether ductile or brittle with changing strain rates?   Is it possible to share pictures of the fractured specimen?  

2) In section 4.4 and the conclusion, emphasis is given to the impact of different parameters on mechanical properties that can help design polymer-based propellants of desired mechanical properties. However, the validation presented in the articles only shows stress-strain curves at different strain rates and it is difficult to guess what is the performance of the model in predicting the impact of other parameters on mechanical properties. A detailed analysis and sound scientific evaluation of the impact of other parameters e.g. plasticizer, binder content etc. is necessary to highlight the robustness of the model. Is there such kind of experimental data that can be used to compare the impact of individual components of material on its mechanical behaviour for a given strain rate? The validation of the impact of other parameters on mechanical behaviour based on experimental data is missing only the impact of different parameters as revealed by the model is given. 

The are some minor corrections which are recommended as below. 

1) Please use a standard citation method for citing references. e.g. Use only last names. Page 3 instead of Zou Z et al use Zou et al. [ ] and also write a reference with the name. Also where citations are given e.g. [16] please ensure a space.  For multiple reference use [28-31] or as per journal guide. 

2) Some typos are there in superscript and subscript e.g. Page No. 5 mmol.g-1, -1 should be in superscript. Also on 12 in R2, 2 should be in superscript. This correction needs to be done in the whole manuscript. 

3) Please check all equations are properly cited in the text. For example, Equation (1) shows .....etc. There is no equation (2), please check.  

Author Response

Comments 1: An interesting phenomenon is visible with an increasing strain rate that shows necking in the test zone of the specimen while at a low strain rate, a brittle failure seems to happen. Can authors comment on the failure mode whether ductile or brittle with changing strain rates?   Is it possible to share pictures of the fractured specimen?

Response 1:  â‘  We greatly appreciate the valuable comments from the reviewer.

Regarding your question about how to comment on the failure mode whether ductile or brittle with changing strain rates. At low strain rates, HTPE propellant typically undergoes ductile failure. However, as the strain rate increases, the failure mode gradually transitions to brittle failure. When the stress reaches a critical value, the material will fracture suddenly with almost no plastic deformation.

This is because the binder in the HTPE propellant has good visco-elastoplasticity. At low strain rates, the specimens typically exhibit typical ductile failure characteristics. As the strain rate increases, the failure mode of the specimen may transition from ductile fracture to brittle fracture. This is because, at low strain rates, the crack propagation speed is relatively slow, and the material has the opportunity to relieve the stress concentration at the crack tip through plastic deformation, thus suppressing the rapid propagation of the crack. Meanwhile, at low strain rates, the specimen has sufficient time to adjust its internal structure and rearrange its molecular chains, thereby achieving plastic deformation of the material. The movement and interaction of dislocations can consume energy and delay the crack propagation, enabling the material to exhibit good toughness.(ref : Eşiyok H. A study on hydroxyl terminated polyether based composite propellants[J]. 2016.)

Under high - strain - rate conditions, the HTPE propellant exhibits typical brittle failure characteristics. This is because high strain rates accelerate the crack propagation speed. The material does not have enough time to relieve the stress concentration at the crack tip through plastic deformation, and the crack will rapidly expand, leading to material fracture. At high strain rates, the movement of dislocations within the material is restricted, and they cannot fully proliferate and move. This reduces the material's ability to undergo plastic deformation and makes it more prone to brittle fracture. (ref : Murri W J, Curran D R, Seaman L. Fracture model for high energy propellant[C], AIP Conference Proceedings. American Institute of Physics, 1982, 78(1): 460-464.)。

The following section is added in line 387 to line 390 in Page 14.

At low strain rates, HTPE propellant typically undergoes ductile failure. However, as the strain rate increases, the failure mode gradually transitions to brittle failure. When the stress reaches a critical value, the material will fracture suddenly with almost no plastic deformation.

â‘¡ Figure 1 and Figure 2 show the macroscopic morphologies of the fractured specimens of HTPE propellants with Formulation 1 and Formulation 2 respectively, under different tensile rates. As can be seen, the positions where the fracture surfaces of the specimens occur at different strain rates are not regular. However, the fracture surfaces are perpendicular to the tensile direction, indicating that the uniaxial tensile fracture of HTPE propellant follows the maximum normal stress criterion (the first strength theory or the maximum tensile stress theory). This criterion holds that the fracture of a material is caused by the maximum tensile stress, that is, when the maximum tensile stress reaches a certain critical value, the material will fracture.

Figure 1. Macroscopic morphology of formulation 1 specimens at different tensile rates.

Figure 2. Macroscopic morphology of formulation 2 specimens at different tensile rates.

Comments 2: In section 4.4 and the conclusion, emphasis is given to the impact of different parameters on mechanical properties that can help design polymer-based propellants of desired mechanical properties. However, the validation presented in the articles only shows stress-strain curves at different strain rates and it is difficult to guess what is the performance of the model in predicting the impact of other parameters on mechanical properties. A detailed analysis and sound scientific evaluation of the impact of other parameters e.g. plasticizer, binder content etc. is necessary to highlight the robustness of the model. Is there such kind of experimental data that can be used to compare the impact of individual components of material on its mechanical behaviour for a given strain rate? The validation of the impact of other parameters on mechanical behaviour based on experimental data is missing only the impact of different parameters as revealed by the model is given.

Response 2: 

We sincerely appreciate your valuable comments on our paper. Regarding the issues you pointed out, we are fully aware of the deficiencies in model validation and parameter influence analysis in our paper. To more comprehensively validate the model's ability to predict the effects of different parameters on the mechanical properties of HTPE propellants, we incorporated existing data and added predictions of the stress-strain curves of HTPE/HATO/AP/Al propellants under different coarse-particle-size AP contents. We used three models, namely the Feedforward Neural Network (FFNN), Kolmogorov-Arnold Networks (KAN), and Long-Short-Term Memory (LSTM) networks, for the predictions. The R2 values of FFNN, KAN, and LSTM on the test set are 0.972, 0.979, and 0.998 respectively. The prediction performance results of the three models have been added to Table 5 (There is an addition to page 18, lines 478). The LSTM model shows better accuracy and stability in predicting the influence of the coarse-particle-size content of AP on mechanical properties, further demonstrating the reliability of the model when dealing with such parameters. We obtained a series of prediction results of stress-strain curves for HTPE/HATO/AP/Al propellants at different coarse-particle-size AP contents through the LSTM model. These results are presented in the paper as Figure 14 “Figure 14. Results of using LSTM models to predict stress-strain curves for HTPE/HATO/AP/Al propellants at different test temperatures for different content of coarse-particle-size AP.”. As can be clearly seen from the figure, with the change in the coarse-particle-size content of AP, the stress-strain curves show an obvious changing trend. At low temperatures, as the coarse-particle-size content of AP increases, the tensile strength first rises rapidly and then the growth rate slows down. This is reflected in the stress-strain curve as the peak value of the curve gradually increases and the upward slope is larger in the early stage and smaller in the later stage. This not only visually shows the influence of the coarse-particle-size content of AP on the mechanical properties but also further validates the effectiveness of the model in predicting the influence of such parameters on mechanical properties.

The part is added from page lines 517-533 of page 21.

In Figure 14, the LSTM model predicts the stress-strain curves of HTPE/HATO/AP/Al propellants under different tensile temperatures and different coarse-particle-size AP contents, with an R2 value reaching 0.998. The LSTM model captures the influence trends of the coarse-particle-size AP content on the tensile strength and elastic modulus at different test temperatures. Especially in a low-temperature environment, the model reveals that both the tensile strength and the elastic modulus increase with the increase in the coarse-particle-size AP content, but this increase effect gradually diminishes. At low temperatures, the mobility of molecular segments in the elastomer matrix is significantly weakened, and the coarse-particle-size AP undertakes more tensile loads, enhancing the material's rigidity. However, an increase in the content of coarse-particle-size AP can lead to uneven distribution and aggravated stress concentration, restricting the enhancement effect and causing the increase in tensile strength and elastic modulus to gradually decrease as the content of coarse-particle-size AP increases.

Figure 14. Results of using LSTM models to predict stress-strain curves for HTPE/HATO/AP/Al propellants at different test temperatures for different content of coarse-particle-AP.

And in accordance with the reviewer's comments, we have enriched the corresponding conclusions in lines 615 - 617 on page 24 of the Conclusion section.

Comments 3: Please use a standard citation method for citing references. e.g. Use only last names. Page 3 instead of Zou Z et al use Zou et al. [ ] and also write a reference with the name. Also where citations are given e.g. [16] please ensure a space.  For multiple reference use [28-31] or as per journal guide.

Response 3:  We are grateful for the comment of the reviewers. In accordance with the your comments and the journal guide, we have revised all the references cited in the full text using standard citation methods.

Comments 4: Some typos are there in superscript and subscript e.g. Page No. 5 mmol.g-1, -1 should be in superscript. Also on 12 in R2, 2 should be in superscript. This correction needs to be done in the whole manuscript.

Response 4: Thanks for your comments. All typos in the superscripts and subscripts have been corrected in the text on a case-by-case basis in response to comments.

Comments 5: Please check all equations are properly cited in the text. For example, Equation (1) shows .....etc. There is no equation (2), please check.  

Response 5: Thank you for pointing this out. We have checked all equations cited in the text and addressed missing formulas.

Reviewer 2 Report

Comments and Suggestions for Authors

This manuscript is an important contribution using the deep neural networks to explore the constitutive relationship of composite solid propellants based on uniaxial tensile experiments of HTPE propellants. In recent years, there have been an increase in the synthesis of energetic materials and new tool are needed to determine their properties. The strategy accurately predicts mechanical behavior of materials with limited data. It can analyze complex composition-property relationships, and reduces  the need for expensive, hazardous, and time-consuming physical experiments. I recommend accepting the manuscript after addressing the minor issues.

Line 18, Describe FFNN, KAN, and LSTM in the abstract.

Line 25, Describe SHAP  and AP.

Line 34, “All six insensitive performance evaluation tests” some information can be provided about the test such as insensitive toward impact, friction, etc.

Page 11. Figure 3 should be Figure 5.

Table 1. Describe the % of HTPE propellant in the formulation

Figure 1, text in the figure 1 is small and hard to read.

Comments on the Quality of English Language

no comments

Author Response

Comments 1: Line 18, Describe FFNN, KAN, and LSTM in the abstract.

Response 1: Thank you for pointing this out. FFNN, KAN, and LSTM have been described in the line 18 of the abstract as you suggested.

Comments 2: Line 25, Describe SHAP  and AP

Response 2: Thank you for pointing this out. SHAP and AP have been described in the line 25 of the abstract as you suggested.

Comments 3: Line 34, “All six insensitive performance evaluation tests” some information can be provided about the test such as insensitive toward impact, friction, etc.

Response 3: We appreciate the valuable comments from the reviewers. The Hydroxyl-Terminated Polyether (HTPE) propellant has passed all seven insensitive performance evaluation tests of the US Military Standard MIL-STD-2105C. These tests are Safe Drop, Fast Cook-off, Sympathetic Detonation, Fragment Impact, Bullet Impact, Slow Cook-off, and Jet Impact. The purpose is to ensure that when ammunition is subjected to accidental stimuli (such as heat, shock, friction, etc.), the risk of accidental explosion or deflagration can be minimized.

The following section contains test information for several insensitive performance evaluation tests.

The purpose of the Safe drop test is to simulate accidental drop scenarios that ammunition may encounter during transportation, storage, and handling. This test assesses the ammunition's impact resistance and determines whether accidental initiation, explosion, or other hazardous situations will occur upon dropping.

The purpose of the sympathetic detonation sensitivity test is to evaluate the insensitivity of materials when subjected to the influence of an adjacent explosive source. That is, it determines whether the explosion of one explosive can trigger the explosion of adjacent explosives of the same or different types.

The follow part is added in line 37 to line 40 in Page29.

The hydroxyl-terminated polyether (HTPE) propellant has passed all seven insensitive performance evaluation tests (Safe Drop, Fast Cook-off, Sympathetic Detonation, Fragment Impact, Bullet Impact, Slow Cook-off, and Jet Impact) of the U.S. Military Standard MIL-STD-2105C.

Comments 4: Page 11. Figure 3 should be Figure 5.

Response 4: Thank you very much for pointing this out. We have changed Figure 3 to Figure 5 according to your suggestion.

Comments 5: Table 1. Describe the % of HTPE propellant in the formulation

Response 5: Thank you very much for your comments, the percentage of each component of HTPE propellant for both formulations has been written in Table 1.

Table 1. HTPE propellant formulation information.

HTPE

Bu-NENA

N-100

Al

AP (6-8μ)

AP (100-140μ)

TPB

HX-878

Formulation 1

9.93%

12.9%

2.02%

10%

15%

50%

0.05%

0.1%

Formulation 2

9.93%

12.9%

2.02%

10%

20%

45%

0.05%

0.1%

Comments 6: Figure 1, text in the figure 1 is small and hard to read.

Response 2: Thank you very much for your comments, the text in Figure 1 has been made larger as per your suggestion. 

Figure 1. (a) HTPE propellant preparation process. (b) Schematic diagram of HTPE propellant uniaxial tensile experiment and specimen (unit: mm)

Response to Comments on the Quality of English Language

Point 1: English of the manuscript needs polishing

Response 1:  We tried our best to improve the manuscript and made some changes to the manuscript. These changes will not affect the content and framework of the paper. We have broken down the long and complex sentences in the article to make the content more understandable. We sincerely appreciate your dedicated work and hope that the revised version will meet with your approval.
